# The Candidate Antigens to Achieving an Effective Vaccine against *Staphylococcus aureus*

**DOI:** 10.3390/vaccines10020199

**Published:** 2022-01-27

**Authors:** Hamid Reza Jahantigh, Sobhan Faezi, Mehri Habibi, Mehdi Mahdavi, Angela Stufano, Piero Lovreglio, Khadijeh Ahmadi

**Affiliations:** 1Animal Health and Zoonosis, Department of Veterinary Medicine, University of Bari, 70010 Bari, Italy; angela.stufano@uniba.it; 2Interdisciplinary Department of Medicine, Section of Occupational Medicine, University of Bari, 70010 Bari, Italy; piero.lovreglio@uniba.it; 3Medical Biotechnology Research Center, School of Paramedicine, Guilan University of Medical Sciences, Rasht 41937, Iran; Faezi@gums.ac.ir; 4Department of Molecular Biology, Pasteur Institute of Iran, Pasteur Ave., Tehran 13164, Iran; m_habibi@pasteur.ac.ir; 5Advanced Therapy Medicinal Product (ATMP) Department, Breast Cancer Research Center, Motamed Cancer Institute, Academic Center for Education, Culture and Research (ACECR), Tehran 1517964311, Iran; 6Recombinant Vaccine Research Center, Faculty of Pharmacy, Tehran University of Medical Sciences, Tehran 13164, Iran; mmahdavi@nanochelatingtechnology.com; 7Infectious and Tropical Diseases Research Center, Hormozgan Health Institute, Hormozgan University of Medical Sciences, Bandar Abbas 79391, Iran

**Keywords:** vaccine, staphylococcus aureus, antigen, virulence factors

## Abstract

*Staphylococcus aureus* (*S. aureus*) is an opportunistic pathogen that causes various inflammatory local infections, from those of the skin to postinfectious glomerulonephritis. These infections could result in serious threats, putting the life of the patient in danger. Antibiotic-resistant *S. aureus* could lead to dramatic increases in human mortality. Antibiotic resistance would explicate the failure of current antibiotic therapies. So, it is obvious that an effective vaccine against *S. aureus* infections would significantly reduce costs related to care in hospitals. Bacterial vaccines have important impacts on morbidity and mortality caused by several common pathogens, however, a prophylactic vaccine against staphylococci has not yet been produced. During the last decades, the efforts to develop an *S. aureus* vaccine have faced two major failures in clinical trials. New strategies for vaccine development against *S. aureus* has supported the use of multiple antigens, the inclusion of adjuvants, and the focus on various virulence mechanisms. We aimed to present a compressive review of different antigens of *S. aureus* and also to introduce vaccine candidates undergoing clinical trials, from which can help us to choose a suitable and effective candidate for vaccine development against *S. aureus*.

## 1. Introduction

*Staphylococcus aureus* (hereafter called *S. aureus*) belongs to the Micrococcaceae family and appears as Gram-positive cocci, 1 μm in diameter, in grape-like clusters. *S. aureus* detection method from other Staphylococcal species is based on the gold pigmentation of colonies and positive results of coagulase, mannitol fermentation, and deoxyribonuclease tests [1]. This microbe is a commensal bacterium, and the nasal carriage of *S. aureus* in the human population is about 30%. This organism also is a ubiquitous human pathogen and the most common cause of skin and soft tissue infections (SSTIs) and endocarditis. *S. aureus* is a predominant cause of a variety of nosocomial infections, including ventilator-associated pneumonia, intravenous catheter-associated infections, postsurgical wound infections, as well as invasive infections in immunosuppressed patients [2,3,4,5]. In the last three decades, Staphylococcal infection has become one of the most common causes of post infection glomerulonephritis [6]. Use of a central venous catheter (CVC) increases the risk of sepsis caused by *Staphylococcus aureus*. The hemodialysis vascular access is a potential entry site for *S. aureus* [7]. *Staphylococcus* infections can cause diseases due to direct infection or by the production of toxins by bacteria such as in food poisoning and toxic shock syndrome. SSTIs (abscess, furuncles, carbuncles), bacteremia (bloodstream infection), infective endocarditis (native and prosthetic valves), pneumonia, osteomyelitis, septic arthritis, central nervous system infections, septic thrombosis of cavernous or dural venous are all examples of direct infection caused by *S. aureus* [8]. Anyone could be affected by *S. aureus* infection, but certain groups of people have a higher risk for developing these infections, including patients receiving hemodialysis, intravenous drug users, patients with diabetes, and patients with preexisting cardiac conditions or other comorbidities. In a healthcare setting, immunocompetence patients or engaged in processes such as surgery or intravenous catheters are at high risk of *S. aureus* infection [9]. Methicillin-resistant *S. aureus* (MRSA) strains are resistant to many antibiotics, even against the antibiotics that are approved for the treatment of the *S. aureus* infections such as linezolid, daptomycin, vancomycin, and clindamycin. Methicillin was first introduced in 1959 in the UK to prevent the growth of *S. aureus* resistance to penicillin. Two years after its introduction as an antibiotic, certain methicillin-resistant cases were reported [10]. The main mechanism of resistance to methicillin in *S. aureus* is through the *mecA* gene that encodes a 76 kDa penicillin-binding protein (PBP2A), with decreased affinity for β-lactam antibiotics [11,12,13,14,15,16]. β-lactams bind to the PBP, which is essential for cell wall biosynthesis and inhibition of peptidoglycan crosslink formation, leading to bacterial cell lysis. Therefore, reduced affinity to β-lactam antibiotics can lead to resistance to a diverse range of β-lactam antimicrobial agents such as methicillin [17].

MRSA infections are common in both adults and children. The average proportion of MRSA in the EU has large intercountry variations, from less than 1% in Denmark, Iceland, Norway, and Sweden to more than 25% in other countries in 2007. MRSA is highly prevalent in hospitals worldwide. High rates (>50%) of MRSA were reported in Asia, Malta, North, and South America in the early 2010s [16]. Asia has the most prevalence of hospital-acquired MRSA (HA-MRSA) and community-associated MRSA (CA-MRSA) in the world.

Mortality, morbidity, and the cost of invasive *S. aureus* infection has control caused the introduction of several new antibiotics to treat MRSA infections, leading to increased costs for health-care services. *S. aureus* infections are now the most common cause of hospitalization for the surgical drainage of pus in children and bacteremia in persons aged >65 years, and the most serious cause of prosthetic device and intravascular line infections [15,18,19,20,21]. The total annual cost of SSTI hospitalizations between 2001 and 2009 increased 26%, implying an increase in national cost from $3.36 to $4.22 billion [22]. Several economic models consistently suggest that a vaccine against *S. aureus* would be strongly cost-effective when compared to existing conventional treatments. This article summarizes recent developments regarding the pathogenesis of *S. aureus* infections, relying on the well-known virulence factors that may be useful for the development of new vaccine candidates or immune therapeutics.

## 2. Immunity to *S. aureus*

Innate immunity recognizes *S. aureus* by pattern recognition receptors (PRRs) such as Toll-like receptor 2 (TLR2) and nucleotide-binding oligomerization-domain-containing 2 (NOD2), and stimulates the production of antimicrobial peptides and specific cytokine signaling pathways such as IL-1α and IL-1β that promote neutrophil recruitment, which are critical in controlling *S. aureus* infection [23]. Adaptive (acquired) immune responses, which include humoral and cellular immunity, also contribute to the host defense. Humoral immunity is an essential mechanism to reduce the invasion of *S. aureus* [24,25]. The major function of humoral immunity is producing specific antibodies to neutralize the function of virulence factors or to opsonize pathogens to optimize phagocytosis and clearance. It was reported earlier that T-cells are not essential for protection against *S. aureus* in mice, but recent findings have demonstrated that Th1 and Th2-cells can have both beneficial and detrimental roles in *S. aureus* infection [24]. Activation of Th1 leads to the secretion of IFN-γ, which can accelerate the clearance of systemic infection by enhancing macrophage responses and upregulating the expression of MHC molecules. Furthermore, IFN-γ is considered to be a stimulator of immunoglobulin isotype switching to IgG1 and IgG3 antibodies in humans, or homologous IgG2a in mice [25], that can act as an opsonin. Besides, Th2 cells could be activated by Staphylococcal cell wall components, such as peptidoglycan and teichoic acid. Th2 cytokines induce and mobilize antimicrobial peptides such as human β-defensin (HBD)-3 [24]. This highly charged (+11) cationic defense peptide retains activity against *Staphylococci* even at elevated salt concentrations [26]. Given that cytokines produced by Th17 cells such as IL-17A and IL-17F are involved in neutrophil production and recruitment, these cells play an essential role in the primary defense against *S. aureus* infections. Th17-associated immune responses can be targeted for strategies to mitigate distal infections originating from persistent *S. aureus* carriage in humans [27].Considering that one of the reasons for the failure of the V710 vaccine against *S. aureus* and increased mortality of vaccine recipients was low serum IL-2 and IL-17A concentrations, it can be concluded that IL17a is critical in eradicating *S. aureus* in the host [28].

## 3. *S. aureus* Vaccine Development

Successful vaccines have been developed against various microbial diseases. Today, diphtheria and tetanus are preventable by toxoid vaccines. *Neisseria meningitidis*, *Haemophilus influenzae B,* and many *Streptococcus pneumoniae* infections are prevented by capsule vaccines. Rubella, measles, mumps, polio, and smallpox vaccines prevent viral infections. The success of these vaccines led researchers to believe that targeting virulence factors in organisms such as *S. aureus* can also lead to effective vaccines. However, investigators believe that vaccination against *S. aureus* would be more complex [29,30].

So far, many studies have been conducted to find an appropriate vaccine against *S. aureus* (Table 1).

StaphVAX was a bivalent polysaccharide vaccine candidate that includes the two most prevalent capsular polysaccharides, CP5 and CP8, conjugated to the detoxified form of *Pseudomonas aeruginosa* exotoxin A. CP5 and CP8 are associated with approximately 80% of *S. aureus* clinical infections. The vaccine has completed phase II clinical trials in chronic, ambulatory, and peritoneal dialysis patients and was safe and effective. Phase III trial was evaluated in patients who were candidates for cardiovascular surgery in the 3–54 weeks following immunization. Initial studies of phase III of the vaccine were positive, but the FDA indicated that US registration would require a second phase III trial following preliminary results. Results of the second phase III studies showed no reduction in *S. aureus* infections in the StaphVAX group when compared to the placebo group. StaphVAX was shown to reduce *S. aureus* bacteremia by 64% through 32 weeks follow-up, by 57% through 40 weeks, and by 26% by 54 weeks. Thus, the antibody titer decreases after 32 weeks. Staphvax failed in the second phase III trial in hemodialysis patients [25,31,32].

The V710 vaccine contains the iron surface determinant B (IsdB), a highly conserved *S. aureus* surface protein developed by Merck and Intercell. Primary studies of the V710 vaccine showed good immunogenicity in animal challenge models. Between 2007 and 2011, the V710 vaccine was evaluated in clinical trial studies. In 2011, the Phase IIb/III of the trial was initiated to evaluate the safety and efficacy of preoperative vaccination in patients undergoing cardiothoracic surgery. Among those with median sternotomy, V710 vaccine use, as compared with placebo, did not reduce the rate of serious postoperative *S. aureus* infections but increased the risk of mortality among the patients. These findings do not support the use of V710 vaccine for the patients undergoing surgical interventions [25,31,33,34].

The studies showed that a reduced level of IL-2 and IL-17 were associated with increased mortality after *S. aureus* infections in V710 vaccine recipients. IL-2 stimulates Th1 and Th17 cells; therefore, the reduced level of IL-2 can cause an inappropriate cell-mediated immune response and, as a result, increases mortality [34]. Since the IL-17 and IL-2 have a crucial role in clearance and eradicating *S. aureus* in the host, and with the reduction of the level of IL-2 and IL-17 and the increased mortality after *S. aureus* infections in V710 vaccine recipients, this vaccine also failed [35].

### On-Going Clinical Human Vaccine Trials

In 2006, the Vaccine Research International Plc (VRi) company completed phase I clinical trials of SA75 as a vaccine candidate. They used the whole-cell of *S. aureus* that had been killed by chloroform. VRi completed phase I for the treatment of hospital-acquired infections caused by Staphylococcal bacteria. Trial studies indicated that the vaccine was safe for humans in phase I and stimulates immune responses. It appears to be no longer under active development. The problem of the killed vaccines is that they do not consider cellular immunity and only antibody responses are measured, without determining their functionality or measuring T-cell responses [36].

Pfizer’s SA4Ag (PF-06290510) candidate comprised four antigens: the adhesion molecule ClfA, the manganese transporter MntC, and anti-phagocytic capsular polysaccharides five and eight conjugated to CRM197. Results of phase I/II demonstrated the safety, tolerability, and immunogenicity of the SA4Ag vaccine. This combination is designed to elicit robust humoral and cellular immune responses against different virulence factors that are essential for the survival of bacteria. The rise in functional antibody titers against *S. aureus* was maintained through at least 12 months. In phase IIb, SA4Ag was also safe and efficient in adults undergoing elective spinal fusion surgery and is currently underway. SA4Ag was granted Fast Track designation by the U.S. Food and Drug Administration (FDA) in February 2014 [31,36,37,38]. Besides, a recent study showed that SA4Ag leads to persistent functional immune responses against *S. aureus* antigens throughout 36 months in healthy adults [39]. Also, further investigation showed that SA4Ag showed an acceptable safety profile and induced rapid and robust functional immune responses in the 20 to 64 and 65 to 85 years groups [40]. This vaccine also showed good results against the progressive condition of *S. aureus* infection in an animal study. SA4Ag vaccination dramatically lowered the bacterial population in deep tissue infection, bacteremia, and the pyelonephritis model. However, these favorable preclinical results with SA4Ag did not show the medical utility of SA4Ag in avoiding surgery-associated, invasive *S. aureus* infection [41].

The GSK (GSK2392103A) vaccine is a four-component Staphylococcal vaccine containing polysaccharides five and eight, conjugated to tetanus toxoid (TT) (CPS5-TT, CPS8-TT), with a mutant form of hemolysin-1 (α-toxin; AT) and ClfA. Tetanus toxoid conjugated to polysaccharide in vaccines leads to the induction of high antibody levels and a robust immune memory response. The phase I completed in 2012 and each vaccine formulation induced robust humoral immune responses after the first vaccine dose [42,43,44,45].

The investigational vaccine, NDV-3, contains the N-terminal portion of the *Candida albicans* (*C. albicans*) agglutinin-like sequence 3 protein (Als3p), formulated with aluminum hydroxide (alum) adjuvant in phosphate-buffered saline (PBS). *C. albicans* Als3p has sequence and structural homology with cell surface proteins of *S. aureus*. Therefore, the NDV-3 vaccine can be effective in both *S. aureus* and *Candida* infections. Phase I completed in 2011, and the safety, tolerability, and immunogenicity of NDV-3 in humans were approved. Phase 2 clinical trials (which are in progress) will evaluate the safety, immunogenicity, and efficacy of the NDV-3A candidate vaccine (NovaDigm Therapeutics, Inc., Grand Forks, ND, USA), to prevent the incidental nasal acquisition of *S. aureus* among a population of military recruits at increased risk for *S. aureus* colonization and disease. NovaDigm Therapeutics completed a phase II of trial in vulvovaginal candidiasis (Prevention, Recurrent) in the USA in 2016 [46,47].

The Novartis four-component *S. aureus* vaccine (4C-Staph) comprises five *S. aureus* antigens: a genetically detoxified derivative of the secreted α-toxin or α-hemolysin (Hla), FhuD2 and Csa1A, and EsxAB (a fusion protein containing EsxA and EsxB). This formulation was able to protect mice from *S. aureus* infection with the induction of specific antibodies. 4C-Staph is in the preclinical phase. Torre et al. found that the vaccination of 4C-Staph in neutropenic mice can lead to the increased recruitment of macrophages and monocytes at the site of infection, and it can compensate the neutrophil deficiency. Neutropenia in humans is one of the pathological conditions that make patients vulnerable to *S. aureus* infections. These findings may have important roles in vaccine development [48,49,50].

STEBVax is a recombinant form of *Staphylococcal* Enterotoxin B (SEB), containing three point mutations (L45R, Y89A, and Y94A) that block the interaction of the toxin with human MHC class II receptors. Phase I has been completed in 2016 and showed strong immunogenicity which induced the generation of specific antibodies. Immunization with SEB protected mice not only against challenges with SEB but also SEA, SEC1, or TSST-1 [51,52,53,54].

After the failure of StaphVAX, Nabi Company resumed development in 2006 and began the development of a modified vaccine called PentaStaph, which consisted of the original StaphVax formulation, as well as teichoic acid, alpha-toxin, and Panton-Valentine leukocidin (PVL). Upon the completion of the Phase I clinical trial, the PentaStaph vaccine was sold to GlaxoSmithKline Biologicals. The PentaStaph vaccine is in phase I/II clinical development [36,55].

**Table 1 vaccines-10-00199-t001:** Summary of the clinical trials of various vaccine candidate antigens against *S. aureus*.

Vaccine Candidate	Antigens	Company	Clinical Trials	Adjuvant	References
**StaphVax**	CP5 & CP8	Nabi	Failed phase III	No adjuvant	[25,31,32]
**V710**	IsdB	Merck	Failed phase III	No adjuvant	[34,35]
**SA75**	Whole cell vaccine	Vaccine Research International	Phase I	No adjuvant	[36]
**SA4Ag**	ClfA, MntC, CP5 & CP8	Pfizer	Phase IIb	No adjuvant	[31,36,37,38,39,40]
**GSK2392103A**	CP5, CP8, tetanus toxoid, mutant forms alpha-hemolysin, and ClfA	GSK	Phase I	AS03B	[41,42,43,44,45]
**NDV-3**	Als3p of the Candida albicans that has sequence and structural homology with Eap, GST-Can, His-Clf on *S. aureus*	NovaDigm Therapeutics	Phase II	Aluminum hydroxide	[46,47]
**4c-Staph**	Hla, FhuD2 and Csa1A, and EsxAB	Novartis	Preclinical	TLR7-dependent	[48,49,50]
**STEBVAX**	SEB	Integrated Bio-Therapeutic	Phase I	Alhydrogel	[51,52,53,54]
**Pentastaph**	StaphVax + wall teichoic acid, PVL (rLukS-PV/rAT) and Hla	GlaxoSmithKline	Phase I/II	No adjuvant	[36,55]

## 4. Virulence Factors of *S. aureus*

The success of microbial vaccines such as those against *Neisseria meningitidis*, *Haemophilus influenzae B*, rubella, measles, mumps, polio, and smallpox led researchers to believe that targeting virulence factors in the organisms such as *S. aureus* can also give rise to effective vaccines [56].Different parts of *S. aureus* such as the capsule, surface proteins, and enzymes are targeted in the vaccine studies designed to protect us against infections [57,58]. The virulence factors of *S. aureus* include antigens, enzymes, and toxins (as shown in Table 2 and Figure 1).

### 4.1. Capsules

Capsule production by *S. aureus* was first described in 1931 by Gilbert. Capsules enhance microbial virulence of the bacterium by making them resistant to phagocytosis [59,60]. Capsular polysaccharides are produced by approximately 90% of clinical isolates. Eleven serotypes of encapsulated strains have been determined. Serotypes one and two produce mucoid colonies on solid medium, and they are rarely encountered among clinical isolates. In contrast to these serotypes (one and two), which are visualized by light microscopy, the microcapsule of the serotypes five and eight can only be visualized by electron microscopy after antibody labeling. The capsular polysaccharide (CPS) of both serotypes are high molecular weight carbohydrate polymers composed of N-acetyl-D-fucosamine, N-acetyl-L-fucosamine, and N-acetyl-D-mannosaminuronic acid. Types five and eight polysaccharides differ only in the linkages between the sugars and in the sites of O-acetylation of the mannosaminuronic acid residues [60,61]. Capsular antigens are one of the first targeted antigens in vaccine studies designed to protect against *Staphylococcal* infections. In recent decades, many studies have been conducted on the efficacy of capsules as candidates for the vaccine. The mechanism of protection by capsular vaccines is due to their role in facilitating the pathogen via opsonophagocytosis [60,62,63,64,65]. Based on many findings, if purified polysaccharides are covalently coupled to protein carrier molecules, antibody levels and T-cell-dependent properties are increased and the antibody level would be kept and stable [31,36,37]. The bivalent CP5/CP8 construct (StaphVax) was investigated in three clinical trials (I, II, and III) in patients in end-stage renal disease receiving hemodialysis. The vaccine evoked lower levels of antibodies than anticipated, but still partially reduced the risk of *S. aureus* bacteremia at certain time intervals within the study. Unfortunately, serum antibody concentrations during phase III after 40 weeks in the vaccinated patients declined. There were no significant differences in the number of deaths in the vaccinated and control groups [60,66,67]. Therefore, StaphVax vaccine failed in phase III clinical trials. Vaccines consisting of *S. aureus* CP5 and CP8 that are now at the stage of clinical trials include SA4Ag, SA3Ag, Pentastaph, and GSK2392103A.

### 4.2. Protein A

Staphylococcal protein A (SpA) is harbored to the cell wall envelope of *S. aureus* and binds the Fcγ domain of immunoglobulin (Ig) and cross-links the Fab domain of VH3-type B cell receptors (IgM). The SpA is known to block opsonophagocytosis and is essential for *S. aureus* to escape from the host immune system. Several studies have implemented SpA as a vaccine candidate against *S. aureus* infection [68].

### 4.3. Adhesins

Adhesins as bacterial cell surface receptors play an important role in the interaction between *S. aureus* and its host cells. The pathogen has a variety of adhesins that attach to different factors of the host, such as human extracellular matrix and plasma proteins. The most common surface adhesins that are covalently bonded to the peptidoglycan cell wall are known as the MSCRAMM (microbial surface components recognizing adhesive matrix molecules) protein family. Well-characterized MSCRAMMs are ClfA and B, Cna, IsdA, B and H, FnBPA and B, and SdrC, D, and E [69,70,71]. The studies have shown that mutant strains of MSCRAMMs are less likely to cause infections in animal models [72].

*S. aureus* produces several proteins that can bind to fibrinogen (Fg), fibronectin, and collagen. Clumping factors A and B (ClfA and ClfB) are MSCRAMMs that covalently link to fibrinogen. ClfA (a 933-amino acid protein) has an A domain at the N-terminal with 519 residues that are exposed on the cell surface, and binds to C-terminal residues of the γ-chain of fibrinogen. Both GPIIb/IIIa receptor of platelet and ClfA bind to the γ-chain of fibrinogen, as a result, ClfA stimulates platelet activation and aggregation by a fibrinogen-mediated, as well as a complement-mediated, mechanism [73,74,75]. Unlike ClfA, which binds to the γ-chain of fibrinogen, ClfB binds to the α chain. ClfB causes platelet aggregation, but little is known about the mechanism by which platelet aggregation occurs. ClfA, ClfB, and the serine-aspartate repeat protein (Sdr) E cause the activation of human platelets in plasma. Studies have shown that ClfA has more potent platelet activation and aggregation than ClfB or SdrE. ClfB and SdrE cause aggregation with longer lag times than ClfA and FnBPA. Also, it was reported that ClfA is also anti-phagocytic and protects bacteria from opsonophagocytosis, which might explain its role as a virulence factor in the infection models of sepsis and arthritis [76]. ClfA is an important virulence factor in several infection models, including endocarditis, sepsis, and septic arthritis [74]. Vaccines containing ClfA that are now at the stage of clinical trials include SA4Ag, SA3Ag, and GSK2392103A.

The Sdr proteins are a type of the MSCRAMM family that are encoded by the tandemly arrayed *sdrC*, *sdrD* and *sdrE* genes, of approximately 2.8, 3.9, and 3.5 kbp, respectively [77]. SdrD protein is required for *S. aureus* abscess formation and is also resistant to immune clearance and systemic disease pathogenesis. SdrE protein can promote platelet aggregation, mediated by binding to a plasma protein that acts as a bridge between the bacteria and a platelet receptor [78]. SdrC protein is involved in the first stage of biofilm formation because of its contribution to strong cellular interactions with hydrophobic surfaces [79].

To adhere to collagenous tissues, *S. aureus* needs a specific receptor. Collagen adhesin (Can) is a cell wall-anchored protein that a domain of this protein is responsible for binding to several types of collagen. Cna is a virulence factor in septic arthritis and osteomyelitis that mediates bacterial colonization of cartilage and bone. There is a correlation between the affinity for collagen and the virulence power [80,81,82].

Iron is a key and necessary nutrient for bacteria. A lack of iron inhibits the growth of invading bacteria and, accordingly, allows the host’s immune system to eliminate the infection [83]. *S. aureus* produces high-affinity iron uptake systems called siderophores, which are secreted small molecules with an extremely high affinity for iron and which out-compete host iron-binding proteins. Iron binds to siderophore uptake by cognate receptors on the bacterial surface, allowing the theft of iron from lactoferrin or transferrin [84]. In 2003, Mazmanian et al. reported that the culture of *S. aureus* in iron deficiency medium can lead to the expression of the genes that are involved in iron acquisition. These series of proteins were named Isd (iron surface determinant), or iron-regulated surface determinant, which comprises six proteins: IsdH or HarA, IsdB, IsdA, IsdC, IsdE, and IsdF. Mazmanian et al. suggested that *S. aureus* acquires iron during infection by the first binding of hemoglobin on the bacterial surface (IsdB). Heme is picked up from hemoglobin (IsdA and IsdB) and transferred to the cell wall (IsdC) and cell membrane by translocation factors (IsdD, IsdE, and IsdF) [85]. Previous studies identified that IsdB could be considered as a vaccine candidate antigen for *S. aureus* infections. V710 vaccine (which contains IsdB) had shown protective efficacy in animal challenge models, but the vaccine failed in phase III of the clinical trials due to safety concerns and low efficacy [33]. This vaccine was used for the patients undergoing cardiothoracic surgery and was compared with placebo. As reported, this vaccine did not reduce the rate of serious postoperative *S. aureus* infections and was associated with increased mortality among patients who developed *S. aureus* infections [86].

*S. aureus* produces two closely-related fibronectin-binding proteins (FnBPs), FnBPA and FnBPB. FnBPs are involved in the pathogenesis of *S. aureus* infection by facilitating attachment of the bacteria to the host cells. Heilman et al. found that FnBPA, rather than FnBPB, was able to adhere to platelets and induce their aggregation and plays a role in the induction of endocarditis caused by *S. aureus* [87]. Evaluation of the FnBP as a vaccine candidate demonstrated that it can provide partial protection against *S. aureus* in a murine model of sepsis [88]. In 2006, Zhou et al. reported that Cna-FnBP is a promising vaccine for the prevention of *S. aureus* infections and that the mice immunized with Cna-FnBP survived significantly longer following the challenge with *S. aureus* than non-immunized mice [89].

*S. aureus* infects tissues rich in elastin and binds to soluble elastin or tropoelastin via the elastin-binding protein of *S. aureus* (EbpS). EbpS is expressed at the cell surface as an integral membrane protein. This protein promotes bacterial attachment to components of the extracellular matrix and plays an important role in the pathogenesis of tissue and wound infection by promoting bacterial adhesion and colonization [90].

Manganese transport protein C (MntC) is a highly conserved protein among MRSA and VRSA strains, which has been shown to confer protective immunity in animal model systems of *S. aureus* infections as a vaccine candidate [89,91,92]. MntC significantly increases serum IgG levels and induce cellular immunity. It is a cofactor for superoxide dismutase, which neutralizes superoxide radicals generated during the oxidative burst in the phagosome of macrophages and neutrophils. *S. aureus* strains that lack functional MntC display increased sensitivity to superoxide radicals [93]. MntC is a component of the SA4Ag vaccine in Phase II clinical trials [94].

*S. aureus* uses a specialized ESS pathway to secrete proteins. The pathway which is under the control of *ess* locus encodes the type VII secretion system (T7SS). ESS pathway secretes proteins (EsxA, EsxB, EsxC, and EsxD) during *Staphylococcal* infection. Expression of *ess* genes is required for the establishment of persistent abscess lesions following *S. aureus* bloodstream infection. Staphylococcal Esx proteins mutants that failed to secrete EsxA and EsxB displayed defects in *S. aureus* abscess formation in the mice [95]. EsxA and EsxB are similar to the ESAT-6 and CFP-10 of *M. tuberculosis* and are important during intracellular *S. aureus* infection. Other Ess proteins are important for *Staphylococcal* virulence, such as EsaD, which is located in the Staphylococcal membrane and may contribute to the selection of secretion substrates and/or interact with the Ess secretion machine [95,96,97].

### 4.4. Toxins

Two toxin families enumerated for *S. aureus*, namely (i) Pore-forming toxins, (ii) Superantigens [98].

#### 4.4.1. Pore-Forming Toxins

Toxins that are produced by *S. aureus* target the cytoplasmic membrane and form a pore which leads to the efflux of vital molecules and metabolites out of the cell; therefore, they have a cytolytic activity on the cell membrane. Toxins that lyse red blood cells are called hemolysins, while those that lyse white blood cells are named leukotoxins [99]. These toxins require binding to the receptor for their lytic function.

Alpha-hemolysin (or α-toxin or Hla) has 293 amino acids in length and is expressed by 95% of *S. aureus* strains, as a water-soluble protein of 33 kDa with pore-forming and pro-inflammatory properties [99]. Hla binds to a membrane receptor and forms heptameric pores, which destroys a large range of host cells, including epithelial cells, erythrocytes, fibroblasts, monocytes, macrophages, and lymphocytes, but not neutrophils. The ADAM10 identified as a proteinaceous receptor for α-toxin which is required for toxin binding and oligomerization [100].

Panton-Valentine leukocidin (PVL, consisting of the LukS and LukF proteins) is a toxin produced by *S. aureus* that causes leukocyte destruction and tissue necrosis [101]. It inserts itself into the host plasma membrane and forms a pore. PVL acts synergistically on human and rabbit polymorphonuclear cells (PMNs), monocytes, and macrophages and is the key virulence factor for CA-MRS necrotizing pneumonia and SSTI [102].

Other leukocidins include LukDE and LukAB (LukGH), and gamma-toxins (HlgA, HlgB, HlgC), which have lytic activity on erythrocytes from a wide range of animal species [97,98,101].

*S. aureus* delta-hemolysin (delta-toxin) is a member of the secreted peptides family that are collectively called phenol-soluble modulins (PSMs). These peptides have multiple functions in *Staphylococcal* pathogenesis, such as in lysis of red and white blood cells, controlling biofilm development, and triggering receptor-mediated inflammatory responses. *S. aureus* PSMs have two subfamilies: the peptides with 20–26 amino acids and 43–44 amino acids in length [98,101]. They are small, α-helical amphipathic peptides with detergent-like and pro-inflammatory properties. PSMs attach to the cytoplasmic membrane through non-receptor mediate, leading to membrane disintegration. Pores formed by delta-toxin are short-lived [101,103,104]. *S. aureus* PSMs peptides contribute to the lysis of human neutrophils after phagocytosis, leading to the spread of infection. This can serve as an explanation for the failure of the vaccines based on opsonophagocytosis [105].

#### 4.4.2. Superantigens (SAgs)

Superantigens are a diverse group of protein exotoxins that belong to the most potent T-cell mitogens. With cross-linking between MHC-II and T-cell receptor (TCR) β-chain, they induce the activation of both antigen-presenting cells (APCs) and T lymphocytes, giving rise to the release of large amounts of pro-inflammatory cytokines. Both TCR and MHC-II are contacted outside their antigen-binding sites. SAgs in *Staphylococcus* are members of the Staphylococcal enterotoxins (SEs) family which have a remarkable ability to resist heat and acid. So far, Different types of Staphylococcal enterotoxins have been identified that include A–E, G–J, and R–T (SEA-SEE, SEG-SEJ, SER-SET), t SE-like toxins K–Q and U–X (SElK-SElQ, SElU-SElX), and TSST-1 [98,106,107,108]. SEs are the major cause of food poisoning, which typically occurs by eating foods contaminated with toxins produced by the bacteria *S. aureus* [109,110].

Some strains of *Staphylococcus* produce toxic shock syndrome toxin-1 (TSST-1), which are able to pass through the mucosal surfaces and subsequently to enter into the bloodstream. Also, it increases the lethal effects of endotoxin on renal tubular cells and releases large amounts of cytokines causing high fever, rash, and shock that can lead to death [111,112].

The three *S. aureus* exfoliative toxins ETA, ETB, and ETD are encoded on different genetic elements [98]. They specifically are capable of cleaving the desmosomal cadherin desmoglein 1 (Dsg1), which mediates cell–cell adhesion in a superficial layer of the skin, leading to Staphylococcal scalded skin syndrome (SSSS), a severe skin disease presenting with rash, blisters, and severe lesion damage of the skin. The toxins can spread to the bloodstream. SSSS is rarely fatal in children and does not exceed 5%, but in adults, the mortality rate may reach over 59% despite antibiotic treatment. ETs are superantigens with unique properties that stimulate T cell proliferation [113,114,115].

### 4.5. Enzymes

*S. aureus* can express proteases, lipase, deoxyribonuclease (DNase), and fatty acid modifying enzyme (FAME). It produces four major extracellular proteases: Staphylococcal serine protease (V8 protease; SspA), cysteine protease (SspB), metalloprotease (aureolysin; Aur), and staphopain (Scp). Several in vitro studies have suggested that these enzymes are important virulence factors [116,117,118] for this pathogen. *S. aureus* must change its phenotype from adhesive to invasive due to diffusion within the host. Therefore, bacteria secrete proteolytic enzymes.

SspA cleaves fibrinogen-binding protein and surface protein A (SpA). Aureolysin is responsible for the cleavage of the surface-associated clumping factor, ClfB. SspB is also found as a proteolytically processed product in culture supernatants [118] and induces the rapid engulfment of human PMNs and monocytes by macrophages [79]. Staphopains participate in the development of *S. aureus*-derived ulceration by elastolytic activity and altering the phenotype of the bacteria via cleavage of surface proteins [117,118,119]. Also, the degradation of toxins, such as α-hemolysin, by proteases, and as a result downregulating the virulence of *S. aureus*, help to colonize bacteria on the skin and the nares (*S. aureus* biofilms) [118].

Two *S. aureus* lipases, named SAL1 and SAL2, were identified in *S. aureus* NCTC 8530 (SAL-1) and *S. aureus* PS54 (SAL-2), which degrade triglycerides to release free fatty acids. The lipase interferes with the host granulocyte function and increases the survival of the bacteria during the host defense by inactivating the bactericidal lipids. Clinical strains of *S. aureus* produce lipases that play an important role in biofilm formation, peritoneal abscess formation, and the invasion of the bacteria into host organs [120,121,122]. As a result, lipase could be used as a vaccine candidate against *S. aureus* pathogenesis.

The fatty acid modifying enzyme (FAME) is an extracellular enzyme that inactivates the bactericidal activity of fatty acids by catalyzing the esterification of these lipids to cholesterol, which is not bactericidal for *S. aureus*. About 80% of *S. aureus* produce FAME [123,124,125,126], which, together with lipase, may have a role in determining the survival of *S. aureus* in lesions. In intraperitoneal abscesses, *S. aureus* is destroyed by certain specific lipids produced by the host. The amount of neutral lipids found in *S. aureus* abscesses is much higher than is observed in normal tissues, which FAME inactivates [123,124,125,126].

Adenosine synthase (AdsA) is an extracellular 5′-nucleotidase that catalyzes the conversion of AMP, ADP, and ATP to adenosine. It also converts deoxyadenosine monophosphate (dAMP) into deoxyadenosine (dAdo), which is capable of inducing caspase-3-mediated apoptosis in macrophages and monocytes. The nucleotidase activity of AdsA is critical for *S. aureus* survival in the blood, helps its escape from phagocytic clearance, and causes the formation of abscesses in the organs of infected mice. AdsA increases the concentration of adenosine within the host and suppresses the immune response to escape immune clearance [124,127].

Hyaluronidases are bacterial enzymes that cleave hyaluronic acid (HA) at the β-1, 4 glycosidic bond of hyaluronic acid. This enzyme attacks the interstitial cement of connective tissue by depolymerizing hyaluronic acid and is also involved in biofilm dispersal [128].

Hydrogen peroxide is a powerful and potentially harmful oxidizing agent that can be used to kill bacteria. *S. aureus* produces a catalase enzyme. Catalase can destroy hydrogen peroxide generated during cellular metabolism and neutralizes the bactericidal effects of hydrogen peroxide [129].

Coagulase is an *S. aureus* protein that is best known for its ability to induce blood coagulation by activating prothrombin through the insertion of the Ile1-Val2 N terminus of the Coa D1D2 domain. This enzyme is an important virulence factor in *Staphylococcal* infections that binds to and activates prothrombin, and is required for the formation of abscesses [130].

The penicillinase enzyme hydrolyzes the β-lactam bond of beta-lactam antibiotics and inactive penicillins and cephalosporins. This enzyme is produced by bacteria such as *Bacillus*, *Staphylococcus*, *Escherichia*, and *Klebsiella* [131].

Staphylokinase is an enzyme expressed by *S. aureus*. This enzyme interacts with the host proteins, alpha-defensins, and plasminogen. Staphylokinase enhances bacterial resistance to host innate immunity by interacting with alpha-defensins. Complex binding between staphylokinase and plasminogen reduces the effect of the immune system in *S. aureus* infections and leads to larger lesions with skin disruption [132].

**Table 2 vaccines-10-00199-t002:** The most important virulence factors of *S. aureus*.

References	Current Clinical Trial	Failed Vaccine	Function	Most Important	Antigen
[59,60,61,62,63,64,65]	SA4Ag, SA3Ag, Pentastaph and GSK2392103A.	StaphVax	Polysaccharide	CP5 & CP8	CPs
[87,88,89]			Surface Protein	FnBPA & FnBPB	FnBP
[73,74,75,76]	SA4Ag, SA3Ag and GSK2392103A.		Surface Protein	ClfA & ClfB	Clf
[77,78,79]			Surface Protein	SdrC & SdrD	Sdr
[80,81,82]			Surface Protein	CNA	CNA
[83,84,85]		V710	Surface Protein	IsdB, IsdA	Isd
[90]			Surface Protein		EbpS
[89,91,92,93,94]	SA4Ag		Transporter Protein	MntC	Mnt
[95,96,97]	4c-staph		Extracellular Protein	EsxA& EsxB	ESS
[99,100]	4c-staph		Toxin	Hla	Hla
[101,102]			Toxin	LukS, LukF	PVL
[98,101,102,103,104]			Toxin	Delta Hemolysin	PSM
[113,114,115]			Toxin	ETA, ETB & ETD	exfoliative
[109,110,111,112]	STEBVax		Toxin	Enterotoxins, TSST	SAgs
[116,117,118]			Enzyme	SspA, SspB & Aur	Protease
[128]			Enzyme		Hyaluronidases
[129]			Enzyme		Catalase
[130]			Enzyme		Coagulase
[131]			Enzyme		Penicillinase
[132]			Enzyme		Staphylokinase
[120,121,122]			Enzyme	SAL1 & SAL2	Lipase
[123,124,125,126]			Enzyme	FAME	Fatty acid modifying enzyme
[124,127]			Enzyme	AdsA	Nucleotidase

## 5. Conclusions

Bacterial vaccines have significantly reduced morbidity and mortality caused by several common pathogens, including *Haemophilus influenzae* type B, *Streptococcus pneumoniae*, *Neisseria meningitidis*, diphtheria, pertussis, and tetanus. However, a vaccine to prevent *S. aureus* infections has not been developed yet, despite extensive research studies [15,31]. There are several reasons for justifying the failure of the clinical trials. First, *S. aureus* is part of the normal human flora. It can be found in body parts such as skin, perineum, axillae, vagina, and gastrointestinal tract. Therefore, *S. aureus* has had a long time to adapt to its host environment and its defense system. Second, when compared to other bacterial pathogens, *S. aureus* produces a broader range of virulence factors, including hemolysins, toxins, and superantigens, and it causes different diseases and diseases from cellulitis to food poisoning, osteomyelitis, endocarditis/bacteremia, and prosthetic device infections. Therefore, the type of disease should be considered when designing the vaccine. Third, *S. aureus* is a very complex organism that produces multiple virulence factors, including hemolysins, toxins, and superantigens. Fourth, predictive models cannot be implemented in humans properly. Fifth, the plasticity of the *Staphylococcal* genome means that a large number of strains should be tested before one can have any level of reassurance that the vaccine antigens will be broadly protective [15,133,134]. Therefore, investigators are not only trying to select the most protective antigens but are also working on approaches to counteract the *S. aureus* immune inhibitors. The antibodies against some *Staphylococcal* antigens have created protection in animal models. Thus, the probability of the role of cellular immunity in protection against *S. aureus* infection has been more pronounced and has improved recently in *Staphylococcal* vaccines [14].

The Th1 response accelerates the clearance of systemic infection, thus, Th1 cells represent a novel target for the rational design of future vaccines against *S. aureus* infection. To evaluate a vaccine, antigen selection should be considered in light of both humoral and cellular immunity, and in particular Th17 responses [4,14,37]. Neutralization by specific antibodies against virulence factors has not been established as an effective and protective mechanism against *S. aureus* infections. Therefore, a multivalent vaccine would be more effective, due to its ability to stimulate of humoral and cellular immune responses via antigen variation [37]. Proctor in 2012 suggested that vaccinations based on cellular immunity would lead to better protection against *S. aureus* infections [14]. Overall, the mechanism of immunity for designed vaccines against *S. aureus* infection should be the combination of cellular and humoral immunity, particularly based on the Th17 cell. The Th17 cells produce the effector molecules IL-17, IL-17F, IL-21, and IL-22. IL-17 plays a vital role in the recruitment and activation of neutrophils and may be critical for vaccine-induced memory immune responses against *S. aureus* infections. While the role and ability of the neutrophils is enhanced by antibodies to kill *S. aureus*, the antibody alone without the action of the neutrophils is known to be insufficient to provide protective effects [14,133,135,136]. Recent achievements about the role of Th17/IL-17 in protective immunity are promising in developing vaccines against *S. aureus* infections [137,138]. In our previous study, we designed a recombinant multi-epitope vaccine that elicits high specific IgG titer, and which could also induce a higher level of Th1, Th2, and Th17 cytokines that, for the elimination of *S. aureus* infection, are essential. Also, antisera raised against this recombinant protein indicated a beneficial influence on the opsonophagocytic killing of the bacterial cells that led to a decrease in the bacterial burden in the spleen and kidneys, and might be suitable for potential protection versus *S. aureus* infection [139]. Previous studies have shown that polysaccharides alone are not sufficient to protect against *S. aureus* infection. In a new study, we have conjugated the *S. aureus* capsular polysaccharides five and eight to a fusion protein (Hla-MntCSACOL0723). The results of that study showed that the specific antibody titers against protein polysaccharides five and eight conjugates were higher than the non-conjugated molecules, and conjugated molecules showed stronger humoral immunity [140]. Many studies have shown that the best choice for vaccine design is to employ different antigens in the development of a vaccine [48,141]. All vaccine candidates in clinical phases for *S. aureus* are composed of several antigens. These different antigenic compounds stimulate various immune responses. To simulate various immune mechanisms, multi-epitope proteins that employ immunodominant epitopes of different proteins in the form of a recombinant protein can be utilized [49,141]. For the development of a *S. aureus* vaccine, complex pathogenic mechanisms and numerous pathogenic factors should be considered. In this study, the most important virulent antigens of *S. aureus* that are suitable candidates for the vaccine have been suggested. Each of these antigens described above can activate a specific pathway of the immune system. According to previous studies, a specific combination of these antigens can be introduced as vaccine candidates. Regarding past achievements, the most important criterion for choosing a vaccine candidate should be that it can induce both humoral and cellular immune responses. Therefore, several antigens that target different mechanisms may be the best choice.

## Figures and Tables

**Figure 1 vaccines-10-00199-f001:**
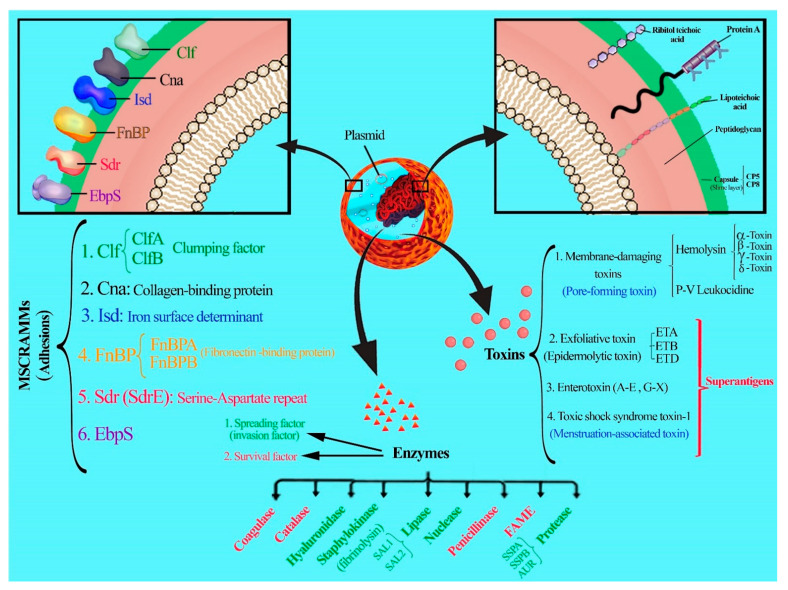
Diverse virulence factors of *S. aureus*.

## Data Availability

Not applicable.

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
