# Peer review of "The Candidate Antigens to Achieving an Effective Vaccine against Staphylococcus aureus"

_vaccines, 2022, doi:10.3390/vaccines10020199_

Round 1

Reviewer 1 Report

The authors present a comprehensive review on vaccine development against Staphylococcus aureus. The topic is of general interest to the readers of the journal. However, some minor issues need to be addressed.

1. Adjuvants are an important component during vaccine development. However, the review article in its current form did not touch upon the adjuvants employed for the development of vaccines against S. aureus. For Table 1, the adjuvants used should be summarized as well.

2. Some minor issues:

1) In Section 4.1, "Capsular polysaccharide (CPS) of both serotypes are is...". Please remove "is".

2) In Section 4.3, the reviewer considers that the word "adhesion" is different from the word "adhesin", whereas, in the section, the two words both appear here and there. Please double-check if you used the right word.

Author Response

Dear reviewer

We thank you for review of our manuscript entitled “The candidate antigens to achieving an effective vaccine against Staphylococcus aureus’’. Now, we believe that the reviewer comments make our manuscript more mature. The comments were answered carefully (point by point) and were applied in the revised revision of the manuscript. The revised parts were highlighted in yellow.

Comment 1: Adjuvants are an important component during vaccine development. However, the review article in its current form did not touch upon the adjuvants employed for the development of vaccines against S. aureus. For Table 1, the adjuvants used should be summarized as well.

Response to comment 1: We much thank the reviewer’s efforts to carefully review the paper and valuable comment. The manuscript revised according to reviewer comment.

Comment 2: 1) In Section 4.1, "Capsular polysaccharide (CPS) of both serotypes are is...". Please remove "is".

Response to comment 2: The suggested correction has been made.

Comment 3: In Section 4.3, the reviewer considers that the word "adhesion" is different from the word "adhesin", whereas, in the section, the two words both appear here and there. Please double-check if you used the right word.

Response to comment 3: The required corrections were made.

Also, The revised manuscript was attached.

Best regards,

Reviewer 2 Report

This is a well written review for current information on S. aureus vaccine

Manuscript is easy to follow and information is adequately provided on the  status of current vaccine application.

There are minor typos which could be fixed by proofreading

I approve to accept as it is 

Author Response

Dear reviewer,

We thank you for review of our manuscript entitled “The candidate antigens to achieving an effective vaccine against Staphylococcus aureus’’. 

The revised manuscript was attached.

Best regards

Reviewer 3 Report

The manuscript "The candidate antigens to achieving an effective vaccine against Staphylococcus aureus" by Hamid Reza Jahantigh and colleagues, is a review article focused on the importance of S. aureus and the current knowledge on virulence factors that may be useful for the development of new vaccine candidates. The study was well conducted, well written, and cite the most relevant studies in the field. The manuscript is very clear, and the conclusions well supported. I have no major concerns about this manuscript.

Minor concerns:

1) Lines 68 and 69, please expand the abbreviation HA and CA.

2) Line 70 “in seven provinces of in collaboration with Professor Alborzi Clinical Microbiology Center”, does not make sense. Please rewrite.

3) The sentence in lines 69-72 (reference 18), is a small and local study that does not bring any relevant information to the global impact of MRSA. Please remove it.

4) Lines 150-151, please elaborate why this vaccine failed.

5) Line 261, please replace spa with SpA.

6) Line 262, please write protein A or (SpA), not both.

7) Line 264, please replace “4.3. Adhesions” by 4.3. Adhesins.

8) Also, in line 268.

9) In table 1 and 2, please add references.

Author Response

Dear reviewer,

We thank you for review of our manuscript entitled “The candidate antigens to achieving an effective vaccine against Staphylococcus aureus’’. Now, we believe that the reviewer comments make our manuscript more mature. The comments were answered carefully (point by point) and all reviewers' comments were applied in the revised revision of the manuscript. The revised parts were highlighted in yellow.

Comment 1) Lines 68 and 69, please expand the abbreviation HA and CA.

Response to comment 1: Thanks for the reviewer’s great attention.  The required corrections were made.

Comment 2) Line 70 “in seven provinces of in collaboration with Professor Alborzi Clinical Microbiology Center”, does not make sense. Please rewrite.

Response to comment 2: The required data was corrected in the revised version of the manuscript.

Comment 3) The sentence in lines 69-72 (reference 18), is a small and local study that does not bring any relevant information to the global impact of MRSA. Please remove it.

Response to comment 3: The required data was corrected in the revised version of the manuscript.

Comment 4) Lines 150-151, please elaborate why this vaccine failed.

Response to comment 4:

Comment 5) Line 261, please replace spa with SpA.

Response to comment 5: The required data was corrected in the revised version of the manuscript.

Comment 6) Line 262, please write protein A or (SpA), not both.

Response to comment 6: The suggested correction has been made.

Comment 7) Line 264, please replace “4.3. Adhesions” by 4.3. Adhesins.

Response to comment 7: The suggested correction has been made.

Comment 8) Also, in line 268.

Response to comment 8: The suggested correction has been made.

Comment 9) In table 1 and 2, please add references

Response to comment 9: We thank the reviewer for this precious comment. The required data was corrected in the revised version of the manuscript.

Also, The revised manuscript was attached.

Best regards
